# The Fitness of Mass Rearing Food on the Establishment of *Chrysopa pallens* in a Banker Plant System under Fluctuating Temperature Conditions

**DOI:** 10.3390/insects12111014

**Published:** 2021-11-11

**Authors:** Jie Wang, Shu Li, Jun Yang, Mingcheng Guo, Huijie Dai, Ricardo Ramirez-Romero, Zhenyu Jin, Su Wang

**Affiliations:** 1Institute of Entomology, College of Agriculture, Yangtze University, Jingzhou 434020, China; wj_insect@126.com; 2Institute of Plant & Environment Protection, Beijing Academy of Agriculture and Forestry Sciences, Beijing 100097, China; lishu@ipepbaafs.cn; 3Institute for the Control of Agrochemicals, Ministry of Agriculture and Rural Affairs, Beijing 100125, China; yangjun2008@agri.gov.cn (J.Y.); guomc90@163.com (M.G.); 4College of Agriculture and Environment, Weifang University of Science & Technology, Shouguang 262700, China; climsion@126.com; 5Biological Control Laboratory, Department of Agricultural Production, CUCBA, University of Guadalajara, Zapopan 317300, Mexico; rramirez@cucba.udg.mx

**Keywords:** banker plant, *Corcyra cephalonica*, lacewing, greenhouse, life table, predation, biological control

## Abstract

**Simple Summary:**

The predatory lacewing, *Chrysopa pallens*, a generalist predator in the field, plays an important role in sustainable, integrated pest management strategies by allowing a reduction in the use of chemical pesticides. However, the effect of mass rearing food, i.e., eggs of the rice moth *Corcyra cephalonica*, on the establishment of *C. pallens* in a banker plant system in the field is unknown. Based on the age-stage, two-sex life table, and predation rate data of *C. pallens* ever cultured on the *C. cephalonica* eggs or the aphid *Megoura japonica* preying on *Aphis craccivora* under fluctuating temperature conditions in a greenhouse, we found that *C. pallens* could complete their development fed on *A**. craccivora* regardless of the food used during culture. This suggests that rice moth eggs could be provided for the mass rearing of predatory lacewings without affecting their population development and biological performance in practical applications compared with lacewings cultured on aphids. This information can serve as a basis for the application of a banker plant system with the mass reared *C. pallens* in the field.

**Abstract:**

Banker plant systems can be used to sustain a reproducing population of biological control agents (BCAs) within a crop, thus providing long-term pest suppression. The founder population of natural enemies in banker plant systems is usually mass-reared on factitious hosts. Thus, a better understanding of the population fitness and pest control performance of mass-reared BCAs in the field is crucial when developing integrated pest management (IPM) strategies. In this study, we determined the fitness of the generalist predator, *Chrysopa pallens* (Hemiptera: Chrysopidae) ever cultured on different food sources (i.e., mass rearing food, *Corcyra cephalonica* eggs, and aphid food, *Megoura japonica*) preying on *Aphis craccivora* in a banker plant system in a greenhouse based on Chi’s age-stage, two-sex life table analysis method. The life tables and predation rate parameters of *C. pallens* were not significantly different between both treatments under fluctuating temperature conditions. *Corcyra*
*cephalonica* eggs did not significantly weaken the performances of *C. pallens* in a *Vicia faba*–*A. craccivora* banker plant system compared to aphids. In conclusion, *C. cephalonica* eggs can be used for the mass production of *C. pallens* as the founder population in a banker plant system. Moreover, linking the life table data with the predation rate is an effective strategy for evaluating mass rearing programs in establishing banker plant systems.

## 1. Introduction

Biological control agents (BCAs), including parasitoids and predators, play an important role in integrated pest management (IPM) strategies that can lead to a reduction in the use of chemical pesticides [1,2,3,4]. Both biocontrol theory and practice suggest that generalist predators can be effective BCAs in IPM [5,6,7]. As a generalist predator, the green lacewing, *Chrysopa pallens* (Rambur) (Hemiptera: Chrysopidae) has been valued for the biological control of pests in agriculture and forestry [8]. It is carnivorous during both the adult and larval stages [9,10] and preys on various pests, including aphids [11], whiteflies, mites [12], and lepidopteran larvae [13]. Research on *C. pallens* has mainly focused on its biology [14,15], investigating the effect of an artificial diet [16,17,18], diapause [14,19], and the detrimental effects of transgenic crops [20,21]. However, knowledge regarding the practical application of *C.*
*pallens* in the field is still far from complete. In particular, the successful establishment of BCA populations released into agroecosystems is challenging and may be heavily reliant on various means to support their populations [22]. Therefore, the development of strategies to support BCA populations could be useful in extending the adoption and efficacy of BCAs in practical applications.

Banker plant systems can be used to sustain a population of reproducing BCAs within a crop, thereby providing long-term pest suppression [23]. By providing shelter and alternative prey/hosts, banker plants can enable the early colonization of these natural enemies [24] and the establishment of their populations when target pests are scarce [25]. Moreover, maintaining the populations of BCAs within an agroecosystem can limit secondary outbreaks of pest populations [26]. Banker plant systems (*Vicia faba–Aphis craccivora*–*C. pallens*) that support *C. pallens* have been developed for pest biological control in commercial greenhouses (Li S, unpublished data), whereas the founder population of natural enemies in banker plant systems is usually mass-reared on factitious hosts [27,28]. Thus, a better understanding of the population fitness and pest control performance of mass-reared BCAs is crucial for developing IPM strategies [1,29].

With the mass application of BCAs, there has been increasing interest in the mass rearing of BCAs on factitious hosts [1,2,4]. Hemipteran herbivores (e.g., *Megoura japonica*) [29], dipteran larvae (e.g., *Ceratitis capitata*) [30], lepidopteran eggs (e.g., *Ephestia kuehniella*, *Sitotroga cerealella*) or larvae (*Musca domestica*) [31,32,33,34], and artificial diets [35,36] are valued as factitious hosts of BCAs. Among them, *Corcyra cephalonica* eggs, a traditional factitious host for mass rearing BCAs, have been well developed for the mass production of parasitoids (e.g., *Trichogramma* wasps) [4] and predators (e.g., *Orius sauteri* [37], *Delphastus catalinae* [38], and predatory lacewings [39,40]). Studies have revealed the suitability of food for the multigenerational mass rearing of BCAs (e.g., parasitoid wasps and predatory mites) [41,42,43]. Nonetheless, only scarce information is available on the adaption of *C. pallens* for mass production in a banker plant system.

The age-stage, two-sex life table, composed of comprehensive datasets regarding the survival, development, and fecundity of a population [44], can precisely delineate stage differentiation, including both sexes in data analysis, description, and interpretation as well as in practical applications [45]. Many studies have used this method to evaluate the effect of temperature, host plant or prey, pesticide, etc. on insects [45,46,47,48]. It is a promising research tool that can accurately assess the field population size and compare the effects of different food sources and environmental conditions on BCAs in pest management. Moreover, linking the life table with the predation rate is also an effective strategy for successful mass rearing programs and the field application of BCAs [45,49,50].

Furthermore, assessing the field population size and stage structure of BCAs is an important topic in pest management and the conservation of beneficial species [45]. In order to facilitate BCAs for biological control, it is valuable to identify the features of populations that are affected by variations in field conditions. Studies have revealed that life history traits are generally affected by fluctuations in the field [51,52,53]. Improving our knowledge of the population characteristics of BCAs in field conditions will be helpful in the mass rearing of insects and their application as natural predators of pests [44].

Therefore, to attain a comprehensive understanding of the fitness of mass rearing food on the establishment of *C. pallens* in a banker plant system, we collected data on the population dynamics and predation rate of *C. pallens* cultured on different food sources (i.e., food for mass rearing or “egg food”, *C. cephalonica* eggs, and “aphid food”, *M. japonica* aphids) in a *V. faba–A. craccivora* banker plant system under fluctuating temperature conditions. We then analyzed the raw data using the age-stage, two-sex life table. Lastly, we discussed the effects of fluctuating conditions on the development and predation rates of BCAs.

## 2. Materials and Methods

### 2.1. Insects

Twenty pairs of adults were collected from Beijing Noah Agricultural Development Co., Ltd. (116°59′ E, 40°6′ N), Beijing, China, in April 2015. For the establishment of the experimental populations for different food treatments, colonies of *C. pallens* were reared on “aphid food” with *M. japonica* and “egg food” with *C. cephalonica* eggs for 10 generations in different custom-made culturing cages (60.0 cm width × 60.0 cm length × 60.0 cm height, constructed using aluminum frames and a plastic fabric 80 mesh net as the walls). In the aphid food treatment, we reared the lacewings with the *M. japonica* food source on broad bean (*V. faba*) with 3–7 true leaves following the method of Cheng et al. [54]. In the egg food treatment, the lacewings were reared following the method of Zhang et al. [55]. The *C. cephalonica* culture was reared at 22–28 °C and 70 ± 5% RH. Fresh eggs were collected daily from rearing plates and exposed to irradiation from an ultraviolet lamp for at least 24 h to kill the embryo. *Chrysopa*
*pallens* were reared with 150 eggs per adult per day, and black papers were provided for oviposition. Either 50 or 100 eggs were provided daily for first- and second- or third-instar larvae, respectively.

*Aphis craccivora* was reared on broad bean. The insects and plants were monitored daily. If necessary, plants were replaced in case of damage caused by aphids. All cultures of *C. pallens* and aphids were kept in air-conditioned rooms at 25 ± 1 °C and 50–70% RH, with a 16:8 h (L:D) photoperiod at the Institute of Plant and Environment Protection, BAAFS, Beijing, China.

### 2.2. Evaluation of Population Colonization of Chrysopa pallens Using Life Tables and Predation Rates

Experiments were conducted in a commercial greenhouse (450 m^2^) from mid-July to late August 2016 (temperature: 26.2 °C average, range 20.6–38.2 °C; relative humidity: range 53–94%) at Beijing Noah Agricultural Development Co., Ltd., (116°59′ E, 40°6′ N), Beijing, China. A total of 100 eggs of *C. pallens* were collected within 24 h from lacewings cultured in each of the two food treatments. The hatched larvae were transferred to individual glass tubes (2 cm in diameter, 7 cm in height) covered with an 80-mesh cotton net within 24 h. Fifty or 100 third- or fourth-instar *A. craccivora* were provided daily to first- and second- or third-instar larvae, respectively, until cocooning. All tubes were placed on a shelf in the greenhouse. Fresh aphids (maintained on broad bean leaf) were supplied daily. The development and survival of each remaining larva and aphid were recorded daily. After the adults emerged, male and female individuals were paired in a cylindrical glass container (8 cm in diameter, 5 cm in height) covered with an 80-mesh cotton net and fed 200 aphids daily. Each day, the survivorship and predation rate of *C. pallens* were recorded, with *A. craccivora* being replaced. When the female adult began to oviposit (about 7 days after eclosion), the female and male adults were separated and placed in individual glass cylindrical containers as mentioned above. A total of 100 aphids were provided daily. The survival, number of eggs laid, and the adult longevity were monitored and recorded daily at regular intervals until the death of the female adult. The longevity, fecundity, and predation rate were recorded daily. The mean daily predation rate per adult was averaged for both sexes because adults were kept as pairs before oviposition.

### 2.3. Life Table Analysis

The development period, survivorship, longevity of individuals, and female daily fecundity of *C. pallens* were analyzed using an age-stage, two-sex life table in the TWOSEX-MSChart program [56,57,58]. The age-stage-specific fecundity ($f_{xj}$, where *x* = age and *j* = stage), age-specific survival rate ($l_{x}$), age-specific fecundity curve ($m_{x}$), preoviposition period of female adults (APOP), total preoviposition period of females from birth (TPOP), and key population parameters (*r*, the intrinsic rate of increase; *k*, the finite rate of increase; $R_{0}$, the net reproductive rate; *T*, the mean generation time) were calculated accordingly.

The age-specific survival rate ($l_{x}$) and age-specific fecundity ($m_{x}$) were calculated as described by Chi and Liu [56] as follows:$$l_{x}=\sum_{j=1}^{\beta }s_{xj}$$
$$m_{x}=\frac{\sum_{j=1}^{\beta }s_{xj}f_{xj}}{\sum_{j=1}^{\beta }s_{xj}}\;.$$

The intrinsic rate of increase (*r*) was calculated using the following formula [59]:$$\overset{\infty }{\underset{x=0}{\sum }}e^{-r\left(x+1\right)}l_{x}m_{x}=1\;.$$

The net reproductive rate ($R_{0}$) was calculated as
$$R_{0}=\overset{\infty }{\underset{x=0}{\sum }}l_{x}m_{x}\;.$$

The mean generation time (*T*) was calculated as *T* = $\ln R_{0}/r$.

The finite rate of increase (λ) was calculated as λ = *e^r^*.

The gross reproduction rate (*GRR*) was calculated using the following formula [60]:$$\mathrm{GRR}=\sum m_{x}.$$

### 2.4. Predation Rate Analysis

The CONSUME-MSChart computer program [61] was used to analyze the predation rates. Following Chi and Yang [57], the age-specific predation rate ($k_{x}$) was calculated as:$$k_{x}=\frac{\sum_{j=1}^{\beta }S_{xj}C_{xj}}{\sum_{j=1}^{\beta }S_{xj}}\;.$$

The age-specific net predation rate ($q_{x}$) was calculated as
$$q_{x}=l_{x}k_{x}\;.$$

The cumulative predation rate ($C_{y}$) was calculated as
$$C_{y}=\overset{y}{\underset{x=0}{\sum }}l_{x}k_{x}\;.$$

The net predation ($C_{0}$) was calculated as
$$C_{0}=\overset{\infty }{\underset{x=0}{\sum }}l_{x}k_{x}\;.$$

The transformation rate from prey population to predator offspring ($Q_{p}$) was calculated as $Q_{p}=C_{0}/R_{0}$.

To compare the predation capacity of a predator on various prey, Yu et al. [30] defined the finite predation rate (ω) as ω=λψ, where *λ* is the finite rate of the predator population and *ψ* is the stable predation rate.

### 2.5. Statistical Analysis

The bootstrap method was used to estimate the standard errors of the developmental time, fecundity, longevity, and population parameters [62] using 100,000 bootstraps in the TWOSEX-MSChart program [58]. The variances and standard errors of $C_{0}$, $Q_{p}$, ω, and λ were estimated using the same 100,000 bootstrap samples from the life table analysis. The bootstrap subroutine is included in the CONSUME-MSChart program [61]. The significance of differences between treatments were calculated using the paired bootstrap test based on the 95% confidence interval in the TWOSEX-MSChart [58] and CONSUME-MSChart programs [61].

## 3. Results

### 3.1. Age-Stage, Two-Sex Life Table of Chrysopa pallens

The developmental time and adult longevity of *C. pallens* were not significantly different between *C. cephalonica* eggs and *M. japonica* treatments under fluctuating temperature conditions in the greenhouse (Table 1). The egg, first-, second-, and third-instar larva, and pupa durations of *C. pallens* were cultured on aphid food (*M. japonica*) was 3.1, 2.7, 3.0, 4.3, and 13.0 days, respectively. When cultured on rice moth eggs, the female *C. pallens* had increased longevity (17.7 days) and males had decreased longevity (7.7 days) compared to those cultured on aphid food. Moreover, no significant differences were found in the APOP, TPOP, and fecundity between both treatments (Table 1).

The age-stage specific survival rates ($S_{xj}$) of *C. pallens* showed the probability of a newborn surviving to age *x* and stage *j*. We can detect stage overlaps in the survival curves because $S_{xj}$ takes into account the variation in individual developmental rates among individuals (Figure 1). The results show that, under fluctuating temperature conditions in a summer greenhouse, the survival rate of *C. pallens* decreased with the developmental stage (Figure 1). The age-stage-specific fecundity ($f_{xj}$) and age-specific fecundity curve ($m_{x}$) initially increased and then decreased with time, showing roughly periodic peaks in reproduction. The maximal daily mean fecundity of *C. pallens* cultured on *C. cephalonica* eggs (4.3 eggs) and *M. japonica* (4.5 eggs) was observed at 42 days (Figure 2).

### 3.2. Population Parameters of Chrysopa pallens under Fluctuating Temperature Conditions in a Greenhouse

The mean and standard errors of the population parameters were estimated using the bootstrap techniques [63]. When exposed to the same prey (*A. craccivora*) and environmental conditions, there were no significant differences in the main population parameters between both treatments. The intrinsic rates of increase (*r*) of *C. pallens* cultured on *M. japonica* and *C. cephalonica* eggs were 0.0379 and 0.0359, respectively (Table 2).

### 3.3. Predation Rate of Chrysopa pallens under Fluctuating Temperature Conditions in a Greenhouse

Under fluctuating temperature conditions in the greenhouse, the predation rate of larva on *A. craccivora* increased with the developmental stage. No significant differences in predation rate were found between both treatments at different developmental stages. The third-instar larvae consumed the greatest amount of prey in the larval stage, reaching 190.8 to 191.1 aphids consumed per larva. Female *C. pallens* lived longer and consumed more aphids than males regardless of the food source (Table 3). A male adult could consume as many as 270 aphids, while a female adult could consume 609 aphids in the *C. cephalonica* egg treatment under fluctuating temperature conditions in the greenhouse.

The daily predation rate of larvae showed the same trend as the age-stage specific predation rate ($c_{xj}$) of *C. pallens* fed on *A. craccivora*, i.e., an initial increase followed by a decrease. The daily predation rate of *C. pallens* cultured on rice moth eggs peaked for the third-instar larvae on the 13th day (55.3 aphids) and was slightly higher than when cultured on aphids (on the 12th day with 53.4 aphids). The nonpredatory stages, including eggs and pupae, were responsible for the two gaps in predation rate (Figure 3). Considering the sex differentiation and stage differentiation, the age-specific predation rate ($k_{x}$) is the mean number of aphids consumed per *C. pallens* of age *x*. Taking the age-specific survival rate ($l_{x}$) into account, the age-specific net predation rate ($q_{x}$) of *C. pallens* can be obtained. With an increase in age, there is a gradual decrease in the age-specific survival rate ($l_{x}$), and less fluctuation in the age-specific predation rate ($k_{x}$). The age-specific predation rate ($k_{x}$) and the age-specific net predation rate ($q_{x}$) curve also exhibited two gaps, representing the sanctuary stage of *A. craccivora* (Figure 4).

The mean and standard errors of the predation rate parameters estimated by means of the bootstrap technique [63] are listed in Table 3. There were no significant differences in the main predation rate parameters between both treatments after incorporating the survival rates and predation rates. The net predation rate ($C_{0}$) of *C. pallens* cultured on aphid food *(**M. japonica*) was higher than when cultured on rice moth *C. cephalonica* eggs. The transformation rates $(Q_{p}$) of *C. pallens* cultured on aphid food (*M. japonica*) and rice moth *C. cephalonica* eggs were 54.22 and 55.97, respectively (Table 3).

## 4. Discussion

The use of banker plants was developed to increase the effectiveness of BCAs in pest control [24,25]. Most studies on banker plant systems have focused on assessing the suitability of a plant species as a banker plant [63,64,65] or the fitness of alternative prey on natural enemies [25,66]. Conversely, studies on the population fitness of BCAs in banker plant systems using alternative prey have received less attention. Yet, with the mass production of BCAs in biocontrol application systems, the population fitness and pest control performance of BCAs in banker plant systems are crucial when developing IPM strategies. The growth rate, stage differentiation and development, fecundity, and predation rate of predatory natural enemies in the field are key to determining their biological control efficiency [67]. The age-stage, two-sex life table is a promising research tool that can be used to accurately assess the effects of different food sources and environmental conditions on BCAs [45]. In the present study, we showed that *C. pallens* could complete its development while preying on *A. craccivora* in a summer greenhouse. Moreover, no significant differences were found in the population and predation rate parameters of *C. pallens* when cultured on either aphid food with *M. japonica* or egg food in the case of *C. cephalonica* eggs. The study indicated that *C. pallens* mass reared on *C. cephalonica* eggs can be suitable for population colonization in a *Vicia faba*–*A. craccivora* banker plant system in IPM.

Laboratory measurements of life parameters usually take place under controlled conditions at single constant temperatures, while field conditions are much more complex with climatic conditions fluctuating over time and space [53,68]. As insects are ectothermic organisms, their responses to constant and fluctuating temperature can vary widely [69,70,71,72,73]. Under controlled conditions in a laboratory, Mu et al. [74] showed that the development of *C. pallens* larvae was 10 to 12 days. Zhao [75] determined the mean developmental times of the egg, larval, and pupal stages of *C. pallens* reared on *A. craccivora* at 25 °C to be 3.38, 11.01, and 13.26 days, respectively, whereas the average durations of the egg, larval, and pupal stages of *C. pallens* reared on *A. craccivora* at 22 °C were 4.3, 11, and 13 days, respectively [30]. Unlike the pupal period, the egg and larval stage durations in our study were shorter than those reported by Yu et al. [30]. Our results are in accordance with the results of previous studies showing that the development time of insects at constant temperature is longer than that under fluctuating temperature conditions [52,76]. On the other hand, the fecundity of female adults in the laboratory was 661 eggs according to Yu et al. [30], which is 18 times higher than that in our studies (33.6–35.5 eggs). Significant reductions in the survival and reproduction of *C. pallens* under greenhouse conditions can be observed in the curves of $S_{xj}$ (Figure 1), $f_{xj}$, and $m_{x}$ (Figure 2). Studies have revealed that the life history traits of insects are generally affected by fluctuations in the field [51,52,53]. For example, in the preadult stage of ladybird *Cheilomenes sexmaculata*, development occurs more slowly under greenhouse conditions, with lower survival and reproductive rates [51]. Our study also revealed different life table parameters for *C. pallens* under laboratory and greenhouse conditions, indicating that the survival rate of *C. pallens* was higher at intermediate temperatures than at high or low temperatures.

Variable temperatures have complex effects on insect performance [77,78]. Under controlled conditions in the laboratory, El-Serafi et al. [79] and Cheng et al. [54] studied the predation rates of *C. pallens* feeding on Aphelinidae (*A. gossypii*, *Sitobion avenae*, *Rhopalosiphum maidis, A. nerii*, and *M. japonica*). The average predation rates of females and males fed on *A. craccivora* in a summer greenhouse in our study were lower than the values determined by El-Serafi et al. [79] and Cheng et al. [54]. Previous studies indicated that fluctuating temperature conditions affect the life history traits of insects, often resulting in low survival rates, fecundity, and longevity [51,78]. Our study also indicated that the predation rate of *C. pallens* was higher at intermediate temperatures than at high or low temperatures. The shorter longevity of adults may have accounted for the lower predation rate under fluctuating temperature conditions.

Mass rearing of BCAs on alternative foods may reduce their performance during the rearing period or after release upon encountering the target prey in the field [80]. Researchers have demonstrated that the natal rearing experience of parasitoids may affect their behavioral and physiological characteristics [80,81,82,83]. Ghaemmaghami et al. [84] found that laboratory mass-reared colonies of *T. brassicae* (Hymenoptera: Trichogrammatidae) declined in quality after 15 generations. On the other hand, the long-term rearing of predatory mite *Neoseiulus californicus* (Acari: Phytoseiidae) on almond pollen positively affected its attributes, including in promoting high survivorship, body size, and fecundity [80]. Our results demonstrate that long-term feeding on mass rearing food, *Corcyra cephalonica* eggs, did not affect the performance of the predatory lacewing *C. pallens* in a *V. faba–A. craccivora*–*C. pallens* banker plant system. Accordingly, mass-reared predatory lacewings can be introduced into a banker plant system for IPM programs.

The specialization of predators on one kind of prey generally entails a tradeoff in performance on another [85]. The net predation rate, stable predation rate, and finite predation rate of *C. pallens* cultured on rice moth *C. cephalonica* eggs and fed on *A. craccivora* were lower than when cultured on aphid food (*M. japonica*). As *C. cephalonica* eggs were stationary when being consumed, *C. pallens* cultured on rice moth eggs had poor performance when exposed to moving prey, i.e., *M. japonica*. More studies need to be conducted to determine the differences in behavior of *C. pallens* cultured on different food. Although the biological performance and population parameters of *C. pallens* cultured on factitious hosts, i.e., *C. cephalonica* eggs, were slightly lower than when cultured on aphid food, there was no significantly negative effect. A life table study incorporating the age-stage predation rate is obviously capable of providing an accurate and thorough understanding of the predator–prey relationship as well as producing a comprehensive evaluation of the potential of a predator/parasitoid as a biological control agent [44,50]. In predator–prey interactions, the nonpredatory egg and pupal stages of the predator represent times of refuge for the prey, and the pest population can grow during these times. Thus, release of *C. pallens* in a mixture of development stages may help to overcome these gaps in a biological control program. That is, natural enemies in banker plant systems can be introduced at mixed stages for better population establishment and pest biological control efficiency. Although the predation rates of *C. pallens* on aphid food observed in our study were lower than those in controlled conditions [54,74,75,79], evidence of the control efficiency of *C. pallens* in practical applications suggests its potential application as a BCA in the field.

Despite an increasing number of studies on biological control programs, there is a paucity of literature on the population and predation parameters of BCAs under variable environmental conditions, even though these are essential in the mass rearing and release of BCAs [51,86]. Moreover, such studies can provide a theoretical basis and practical knowledge for the application of *C. pallens* in greenhouses. Fluctuating temperature conditions as well as a higher humidity, photoperiod, and light intensity may be responsible for the lower survival rate, shorter longevity, and even lower reproduction of predators in the field [51,87]. We suggest that temperature is a key variable that restricts the adult longevity, fecundity, and predation rate of *C. pallens* in the field. The application of banker plant systems in commercial greenhouses require consideration of these fluctuating environments. However, there are many other factors (e.g., light, humidity, population density, and nutrition) affecting the population dynamics and predation ability of *C. pallens*. The susceptibility of insects to numerous environmental factors, natural enemies, and pesticides often varies in relation to their developmental stage [47,88], and more studies regarding the population stage structure under critical conditions are necessary for ensuring effective pest management. Our study confirmed that linking the life table with the predation rate is an effective strategy to evaluate mass rearing programs and the field application of BCAs. Ultimately, these approaches could support the use of generalist predators in more environmentally friendly pest management methods, with reduced negative side-effects compared to regular pesticides in the field [3,89].

## Figures and Tables

**Figure 1 insects-12-01014-f001:**
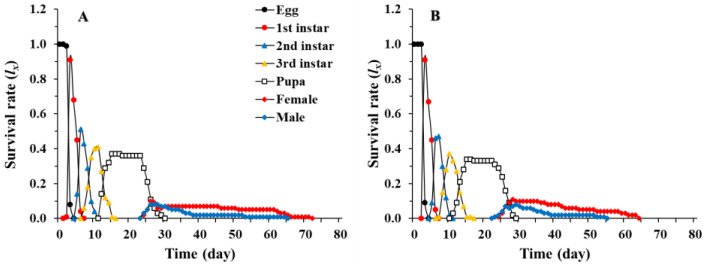
Age-stage specific survival rate ($S_{xj}$) of *Chrysopa pallens* ever cultured on *Megoura japonica* (**A**) and *Corcyra cephalonica* eggs (**B**) under fluctuating temperature conditions in a greenhouse.

**Figure 2 insects-12-01014-f002:**
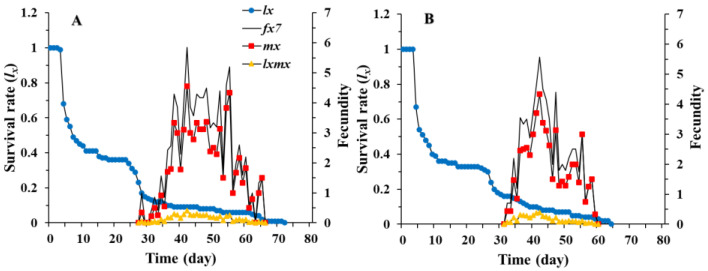
Age-specific survival rate (*lx*), age-stage specific fecundity ($f_{x7}$) of the female stage, age-specific fecundity (*mx*) of *Chrysopa pallens* ever cultured on *Megoura japonica* (**A**) and *Corcyra cephalonica* eggs (**B**) under fluctuating temperature conditions in a greenhouse.

**Figure 3 insects-12-01014-f003:**
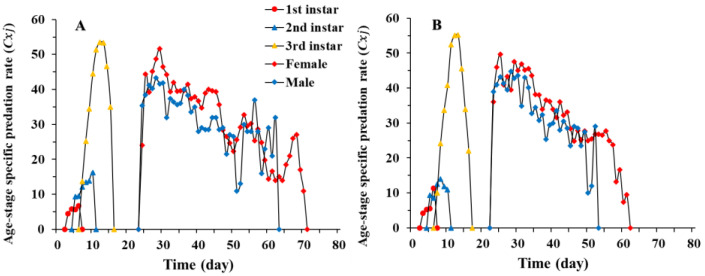
Age-stage specific predation rate ($c_{xj}$) of *Chrysopa pallens* ever cultured on *Megoura japonica* (**A**) and *Corcyra cephalonica* eggs (**B**) under fluctuating temperature conditions in a greenhouse.

**Figure 4 insects-12-01014-f004:**
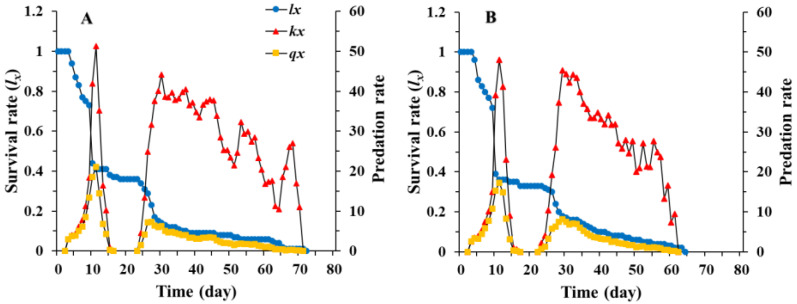
Age-specific survival rate ($l_{x}$), age-specific predation rate ($k_{x}$), and age-specific net predation rate ($q_{x}$) of *Chrysopa pallens* ever cultured on *Megoura japonica* (**A**) and *Corcyra cephalonica* eggs (**B**) under fluctuating temperature conditions in a greenhouse.

**Table 1 insects-12-01014-t001:** Developmental time, longevity, and fecundity of *Chrysopa pallens* ever cultured on *Megoura japonica* and *Corcyra cephalonica* eggs fed on *Aphis craccivora,* under fluctuating temperature conditions in a greenhouse (paired bootstrap test, B = 100,000, the same lowercase letters in the same column indicate the values are not significantly different, *P* > 0.05).

Parameters	Stage	*M. japonica*		*C. cephalonica* Eggs		*P*
		*n*	Mean ± SE	*n*	Mean ± SE	
Developmental time (days)	Egg	100	3.1 ± 0.1 a	100	3.1 ± 0.1 a	0.623
	1st instar	54	2.7 ± 0.1 a	51	2.8 ± 0.1 a	0.080
	2nd instar	45	3.0 ± 0.2 a	42	3.1 ± 0.1 a	0.644
	3rd instar	38	4.3 ± 0.2 a	34	4.2 ± 0.1 a	0.511
	Pupa	30	13.0 ± 0.2 a	28	13.1 ± 0.2 a	0.620
Adult longevity (days)	Female	16	15.8 ± 4.4 a	15	17.7 ± 3.7 a	0.739
	Male	14	8.5 ± 2.7 a	14	7.7 ± 2.3 a	0.820
APOP of female (days)	Female	7	6.6 ± 0.9 a	10	7.3 ± 0.3 a	0.433
TPOP of female (days)	Female	7	33.3 ± 1.0 a	10	33.7 ± 0.4 a	0.710
Fecundity (eggs/female)	Female	16	35.5 ± 14.8 a	14	33.6 ± 11.7 a	0.910

**Table 2 insects-12-01014-t002:** Population parameters of *Chrysopa pallens* ever cultured on *Megoura japonica* and *Corcyra cephalonica* eggs fed on *Aphis craccivora* under fluctuating temperature conditions in a greenhouse (paired bootstrap test, B = 100,000, the same lowercase letters in the same column indicate the values are not significantly different, *P* > 0.05).

Population Parameter	*M. japonica*	*C. cephalonica* Eggs	*P*
Intrinsic rate of increase (r) (day^–1^)	0.0379 ± 0.0119 a	0.0359 ± 0.0109 a	0.916
Finite rate of increase (λ) (day^–1^)	1.0386 ± 0.0122 a	1.0366 ± 0.0112 a	0.916
Net reproduction rate (*R*_0_) (offspring individual^–1^)	5.68 ± 2.64 a	4.67 ± 1.96 a	0.753
Mean generation time (T) (day)	45.83 ± 1.72 a	42.90 ± 1.14 a	0.071
Gross reproduction rate (GRR) (offspring)	72.58 ± 24.29 a	55.61 ± 16.32 a	0.560

**Table 3 insects-12-01014-t003:** Predation rates and parameters of *Chrysopa pallens* ever cultured on *Megoura japonica* and *Corcyra cephalonica* eggs fed on *Aphis craccivora* under fluctuating temperature conditions in a greenhouse (paired bootstrap test, B = 100,000, the same lowercase letters in the same column indicate the values are not significantly different, *P* > 0.05).

Parameter	Stage	*M. japonica*	*C. cephalonica* Eggs	*P*
		Mean ± SE	Mean ± SE	
Predation rate(aphids/predator)	1st instar	14.9 ± 0.9 a	15.1 ± 0.7 a	0.837
	2nd instar	35.5 ± 2.4 a	35.5 ± 2.2 a	0.987
	3rd instar	191.1 ± 5.7 a	190.8 ± 6.3 a	0.954
	Female	539.0 ± 150.0 a	609.0 ± 120.0 a	0.715
	Male	291.1 ± 79.8 a	270.2 ± 68.1 a	0.836
$\mathrm{Net}\;\mathrm{predation}\;\mathrm{rate}\;C_{0}$		236.00 ± 39.66 a	220.97 ± 36.33 a	0.776
Transformation rate $Q_{p}$		54.22 ± 43.93 a	55.97 ± 29.41 a	0.945
Stable predation rate ψ		9.19 ± 0.73 a	8.92 ± 0.78 a	0.799
Finite predation rate ω		9.53 ± 0.80 a	9.23 ± 0.87 a	0.802

## Data Availability

The data presented in this study are available in article.

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
