# Peer review of "The Fitness of Mass Rearing Food on the Establishment of Chrysopa pallens in a Banker Plant System under Fluctuating Temperature Conditions"

_insects, 2021, doi:10.3390/insects12111014_

Round 1

Reviewer 1 Report

Insects-1406605-R1

Title: The fitness of mass rearing food on the establishment of Chrysopa pallens in banker plant system under fluctuating temperatures conditions

Brief.

The manuscript has been improved after revisions. Nonetheless, some corrections are required in the materials and methods. In addition, the critical point remains in the results sections where the results were not described correctly. The manuscript should be polished in standard English.

General comments

L95 – Please, cm3 should be changed to cm. Because each value is referencing a unit on the width x length x height. The only total value would be related to cm3.

L99 – Please, the symbol Ëš for degree Celsius is wrong. Please, the symbol should correctly be inserted in the sentence.

L126 – The ordinal number was written in the begin of sentence in previous sentences. This should be patterned in entire manuscript. Please, 100 should be changed to one hundred in the begin of sentence.

175 to 176 – There were no significant differences of developmental time and adult longevity of C. pallens ever cultured on C. cephalonica eggs and M. japonica under fluctuating temperatures in greenhouse. Please, the sentence should be corrected in the results section.

L213 to 214 – Although the total number of preys consumed by females were visually higher than males of C. pallens, there is no statistical analysis to prove that the total number preys by females were higher than males of C. pallens to two treatments. Please, the sentence should be deleted or rephrased.

L323 to 325 – In the results sections showed that there were no significant differences the net predation rate, stable predation rate, and finite predation rate of C. pallens cultured rice moth C. cephalonica eggs fed on A. craccivora or even cultured on aphids M. japonica (Tables 2 and 3). Please, the sentence should be corrected in the discussion section.

L325 - Please, the contracted form of the English writing (e.g., can’t, etc.) should be avoided in scientific manuscript.

Author Response

Dear academic editor and reviewers,

Thank you for your works on ‘The fitness of mass rearing food on the establishment of Chrysopa pallens in banker plant system under fluctuating temperatures conditions’ (Manuscript ID: insects-1406605). We appreciate your careful feedback and comments in our manuscript. We believe that the manuscript improved based on your contributions. We have taken Reviewers' comments into consideration word by word. And we had added the "Simple Summary" part before the Abstract. We made several modifications based on your suggestions to make clearer the central topic of the manuscript. Please, see our point-by-point responses to your comments and suggestions. We also sent our manuscript to a new round of English language review. We used a professional editing service of MDPI. We believe that the manuscript language improved. More details are provided in the revision with “Track Changes”.

Thank you for considering our revision notes. We are on the way to pest biological control.

Do not hesitate to contact us if you have any questions regarding the revision of our manuscript.

We look forward to hearing from you soon.

Best regard!

Su Wang

BAAFS, Beijing, China

wangsu@ipepbaafs.cn

Comments and Suggestions for Authors

The manuscript has been improved after revisions. Nonetheless, some corrections are required in the materials and methods. In addition, the critical point remains in the results sections where the results were not described correctly. The manuscript should be polished in standard English.

Response: Thanks for your suggestions. We had done some corrections as requested. The results sections were described more correctly according to the reviewer’s comments and suggestions. And our manuscript had checked by a native English-speaking colleague. Please see the revision.

L95 – Please, cm3 should be changed to cm. Because each value is referencing a unit on the width x length x height. The only total value would be related to cm3.

Response: Thanks. We had corrected this sentence into “60.0 cm in width x 60.0 cm in length x 60.0 cm in height”.

L99 – Please, the symbol Ëš for degree Celsius is wrong. Please, the symbol should correctly be inserted in the sentence.

Response: Done as requested. Thanks.

L126 – The ordinal number was written in the begin of sentence in previous sentences. This should be patterned in entire manuscript. Please, 100 should be changed to one hundred in the begin of sentence.

Response: The reviewer is right. Done as requested. Thanks. And we had looked through the manuscript to avoid errors like this.

175 to 176 – There were no significant differences of developmental time and adult longevity of C. pallens ever cultured on C. cephalonica eggs and M. japonica under fluctuating temperatures in greenhouse. Please, the sentence should be corrected in the results section.

Response: Thanks. We had corrected this sentence into “The developmental time and adult longevity of C. pallens had no significant differences between C. cephalonica eggs and M. japonica under fluctuating temperatures in greenhouse”.

L213 to 214 – Although the total number of preys consumed by females were visually higher than males of C. pallens, there is no statistical analysis to prove that the total number preys by females were higher than males of C. pallens to two treatments. Please, the sentence should be deleted or rephrased.

Response: Thanks. We didn’t do the statistical analysis to prove the significance between the two treatments. And we had deleted this sentence.

L323 to 325 – In the results sections showed that there were no significant differences the net predation rate, stable predation rate, and finite predation rate of C. pallens cultured rice moth C. cephalonica eggs fed on A. craccivora or even cultured on aphids M. japonica (Tables 2 and 3). Please, the sentence should be corrected in the discussion section.

Response: Thanks. We had modified this in the discussion section into “In the present study, we showed that C. pallens could complete its development preying on A. craccivora in summer greenhouse. And no significant differences were found in the population and predation rate parameters of C. pallens between Aphid-food” with M. japonica and “Egg-food” with C. cephalonica eggs. The study indicated that C. pallen mass reared on C. cephalonica eggs can be suitable for the population colonization in Vicia faba - A. craccivora banker plant system in IPM.”.

L325 - Please, the contracted form of the English writing (e.g., can’t, etc.) should be avoided in scientific manuscript.

Response: Thanks. We had deleted the “etc.”.

Reviewer 2 Report

Specific Comments attached,

Author Response

Cover letter

Dear reviewer,

Thank you for your works on ‘The fitness of mass rearing food on the establishment of Chrysopa pallens in banker plant system under fluctuating temperatures conditions’ (Manuscript ID: insects-1406605). We appreciate your careful feedback and comments in our manuscript. We believe that the manuscript improved based on your contributions. We have taken Reviewers' comments into consideration word by word. And we had added the "Simple Summary" part before the Abstract. We made several modifications based on your suggestions to make clearer the central topic of the manuscript. Please, see our point-by-point responses to your comments and suggestions. We also sent our manuscript to a new round of English language review. We used a professional editing service of MDPI. We believe that the manuscript language improved. More details are provided in the revision with “Track Changes”.

Thank you for considering our revision notes. We are on the way to pest biological control.

Do not hesitate to contact us if you have any questions regarding the revision of our manuscript.

We look forward to hearing from you soon.

Best regard!

Su Wang

BAAFS, Beijing, China

wangsu@ipepbaafs.cn

Reviewer 2

Line 82 and 250: “Predicted”. Based on the values of life table parameters, did you make any prediction on the population dynamics and predation rate of C. pallens in this manuscript? An illustration missing in the manuscript, when the lab reared C. pallens is released in banker plant scenario in the green house.

Response: Thanks. We were focused on the descriptions on the population dynamics and predation rate of C. pallens in this manuscript. We had deleted “Predicted” for a better understanding. And we firstly want to quantitatively access the fitness of mass rearing food on the establishment of C. pallens in banker plant system under fluctuating temperatures conditions. So, quantitatively fresh aphids were maintained on broad bean leaf in greenhouse. We will evaluate the control effect of C. pallens in potted banker plant scenario in the greenhouse.

Line 84 and 267-269: Did you really compared key parameters (various) in constant temperature with fluctuating temperatures? To support the statement in Lines 267 -269 needs some kind data collection and meta-analysis. In my previous review, I pointed out the authors will need a control treatment to compare, discuss and conclude.

Response: The reviewer has a good point here. First, we are focused on the fitness of mass rearing food on the establishment of Chrysopa pallens in banker plant system under fluctuating temperatures conditions. A control treatment would be useful for quantitively meta-analysis. In this study, we want to understand the change trends of key parameters in constant temperature and fluctuating temperatures. And we discussed roughly the differences between key parameters in constant temperature and fluctuating temperatures with previous studies that described the life parameters/ predation rates of C. pallens in constant temperature (please see Yu et al., 2013; Cheng et al., 2014). We discussed our results with those in constant temperature in the second and third paragraph in the discussion, and we added some statements “Our study also revealed different life table parameters for C. pallens under laboratory and greenhouse conditions, and indicated that the survival rate of C. pallens was higher at intermediate temperatures than at cline margins.”. And we had deleted the “compared” in the Introduction and Discussion parts, based on your suggestions to make clearer the central topic of each paragraph.

Line 262- 265: If this is a conclusion of this manuscript, the research and the manuscript is not justifiable. It is obvious that lacewing heavily feeds on aphids and completes their developmental stages.

Response: That was another good point raised by the reviewer. The previous conclusion of this manuscript is simple and redundant, which didn’t answer the purpose of the research. We changed this paragraph to “In the present study, we showed that C. pallens could complete its development preying on A. craccivora in summer greenhouse. And no significant differences were found in the population and predation rate parameters of C. pallens between Aphid-food” with M. japonica and “Egg-food” with C. cephalonica eggs. The study indicated that C. pallen mass reared on C. cephalonica eggs can be suitable for the population colonization in Vicia faba - A. craccivora banker plant system in IPM.”. We hope that this modification made the text clearer.

Line 86 states “Correlation” between “developmental stages” and “predation rate” discussed and line 267 -271 discussed this statement. This is inadequate. In order to write this statement at the end of the “introduction” section, in the form of the main objective, this needs some evidence discussed based on data analysis.

Response: Linking the life table and predation rate is an essential step in research involving biological control and predator prey relationships. Because the age-stage, two-sex life table does take both sexes and stage differentiation into account, it can precisely describe changes of predation rate that occur with age and stage, and can also include the amount of predation attributable to male individuals. It is obviously capable of providing an accurate and thorough understanding of the predator–prey relationship, as well as producing a comprehensive evaluation of the potential of a predator/parasitoid as a biological control agent (Chi & Yang 2003, Yu et al. 2013, Chi et al., 2020). Besides, “the statements that “Correlation” between “developmental stages” and “predation rate” weren’t the main purpose in our study, and we deleted this. And we added “we discussed the correlation between development and predation rate for assessing the fitness of the traditional mass rearing food, C. cephalonica eggs on rearing BCAs.” to make this paragraph more objective with a primer focus on the fitness of mass rearing food on the establishment of Chrysopa pallens in banker plant system under fluctuating temperatures conditions.

Line 303-304: “significant reductions …….”. Only numbers are smaller, to conclude this need a separate treatment (control) or metanalysis.

Response: Thanks. We didn’t do statistical analysis here. Focused on the trends in the curves of survival, we removed the “significant” to make this paragraph more scientifically.

Line 317 – 318: How long is “long term (feeding) or rearing. How many generations? Line 319: “the predatory lacewing C. pallen in V. faba - A. craccivora banker plant system” is telling me it’s not a result of captivated trial.

Response: Thanks. We had described this in the “Materials and Methods” part: “For the establishment of the experimental populations for different food-treatment, colonies of C. pallens were reared in “Aphid-food” with M. japonica and “Egg-food” with C. cephalonica eggs for 10 generations in different custom-made culturing cages.” We are focused on the fitness of mass rearing food on the establishment of Chrysopa pallens in banker plant system under fluctuating temperatures conditions. So, it is suitable for describing “the predatory lacewing C. pallen in V. faba - A. craccivora banker plant system”. And this results also provided a basis of the performances of C. pallen in field, which to some extent is a result of captivated trial.

The main outcome of this research is “there is no difference found in the population and predation parameters between “aphid-food” and “egg-food”. The most space in this manuscript is taken by illustrating them in terms of text, tables, and figures. Sometimes insignificant data are also critical and should publish so that other researchers would not try again. But this one does not seem to be critical to me. I suggest authors to mention “there is no significant difference between two diets, then choose the best diet (egg diet or aphid diet), present the result and discuss how best it will have BCA implications in the green house. Line 328-321 mentioned no difference but aphid diet is slightly better. 

Response: That was good point raised by the reviewer. There is no significant difference between two diets, and we can indicate the fitness of mass rearing food on the establishment of C. pallens in banker plant system. We had presented the result and discuss how best C. cephalonica eggs will have BCA implications in the greenhouse. We also added the statement that “The mass rearing food, Corcyra cephalonica eggs did not affect the population colonization of C. pallen with Vicia faba - A. craccivora banker plant system compared to aphid treatment. In conclusion, C. cephalonica eggs can be used for mass production of C. pallens as the founder population in the banker plant systems.”.

Reviewer 3 Report

These are my main comments on the MS (insects-1406605) entitled:“ The fitness of mass rearing food on the establishment of Chrysopa pallens in banker plant system under fluctuating temperatures conditions” by Jie Wang and colleagues.

The authors have graphed and presented their results clearly, drawing some attention to the implications of their findings. I found the study of interest and a good contribution to the knowledge of bioecology of BCAs. The methods used are appropriate for the objectives of the work and, in general, well depicted. The resulting figures are sufficient, informative, and of good quality helping to follow the reasoning throughout the manuscript.

The study is however completely divorced from other similar studies on rearing BCAs for field releases. Some of the authors statement would be much stronger if they tie their work to the body of literature that has built up from rearing BCAs across the range of realistic temperatures experienced in the field. They all point to the same direction. Some examples are: J. Econ. Entomol. 112: 1560-1574 and J. Econ. Entomol. 112:1062-1072, but there are others. These studies provide strong evidence of increased longevity in BCAs reared at non-stressful low temperatures when compared to higher temperature regimes. This article should provide details on all these fronts to provide the proper context for the work. They also indicate that the consumption rate of BCAs was significantly higher at intermediate temperatures than at cline margins. Adding these details will improve the paper.

There are still uncertainties about the results obtained, especially because experiments were conducted in a commercial greenhouse at a single temperature regime, ranging from 20C to 38C. Therefore, studies across a broader set of fluctuating temperature regimes are still necessary to understand the real effect of temperature on the characteristics of the BCAs, as this is the closest to the daily temperature fluctuations that occur in the field. This is not to diminish the data gathered in this study, they are of value. But it is important for the authors not to overgeneralize, and to warn the reader, including regulatory agencies, against doing so as well.

Overall, I was excited to see the results of the paper after reading the abstract, but I found it hard to extract key messages useful to policymakers and professionals, probably in large part due to the lack of connection with other published work and need for improved structure of the current manuscript.

Author Response

Cover letter

Dear reviewer,

Thank you for your works on ‘The fitness of mass rearing food on the establishment of Chrysopa pallens in banker plant system under fluctuating temperatures conditions’ (Manuscript ID: insects-1406605). We appreciate your careful feedback and comments in our manuscript. We believe that the manuscript improved based on your contributions. We have taken Reviewers' comments into consideration word by word. And we had added the "Simple Summary" part before the Abstract. We made several modifications based on your suggestions to make clearer the central topic of the manuscript. Please, see our point-by-point responses to your comments and suggestions. We also sent our manuscript to a new round of English language review. We used a professional editing service of MDPI. We believe that the manuscript language improved. More details are provided in the revision with “Track Changes”.

Thank you for considering our revision notes. We are on the way to pest biological control.

Do not hesitate to contact us if you have any questions regarding the revision of our manuscript.

We look forward to hearing from you soon.

Best regard!

Su Wang

BAAFS, Beijing, China

wangsu@ipepbaafs.cn

Reviewer 3

Comments and Suggestions for Authors

The authors have graphed and presented their results clearly, drawing some attention to the implications of their findings. I found the study of interest and a good contribution to the knowledge of bioecology of BCAs. The methods used are appropriate for the objectives of the work and, in general, well depicted. The resulting figures are sufficient, informative, and of good quality helping to follow the reasoning throughout the manuscript.

Response: We appreciate your careful feedback and comments in our manuscript. We believe that the manuscript improved based on your contributions and on the contributions from you. Please, see our point-by-point responses to your comments and suggestions.

The study is however completely divorced from other similar studies on rearing BCAs for field releases. Some of the authors statement would be much stronger if they tie their work to the body of literature that has built up from rearing BCAs across the range of realistic temperatures experienced in the field. They all point to the same direction. Some examples are: J. Econ. Entomol. 112: 1560-1574 and J. Econ. Entomol. 112:1062-1072, but there are others. These studies provide strong evidence of increased longevity in BCAs reared at non-stressful low temperatures when compared to higher temperature regimes. This article should provide details on all these fronts to provide the proper context for the work. They also indicate that the consumption rate of BCAs was significantly higher at intermediate temperatures than at cline margins. Adding these details will improve the paper.

Response: The reviewer made a good point here. We did compare our work to the body of literature that has built up from rearing BCAs across the range of realistic temperatures experienced in the field, but it isn’t systematic. So, we made several modifications in the Discussion, based on your suggestions to make clearer the central topic of each paragraph. And the suggested references are well-known indexes, and we had cited the original articles to provide details on all these fronts to provide the proper context for the work. And another good point here that “the study also indicate that the consumption rate of BCAs was significantly higher at intermediate temperatures than at cline margins.” It’s a good point here, and we had added these details to improve the paper. Thank you. Please see the details in second and third paragraph in the discussion part.

There are still uncertainties about the results obtained, especially because experiments were conducted in a commercial greenhouse at a single temperature regime, ranging from 20C to 38C. Therefore, studies across a broader set of fluctuating temperature regimes are still necessary to understand the real effect of temperature on the characteristics of the BCAs, as this is the closest to the daily temperature fluctuations that occur in the field. This is not to diminish the data gathered in this study, they are of value. But it is important for the authors not to overgeneralize, and to warn the reader, including regulatory agencies, against doing so as well.

Response: The reviewer is right. We clarified the fitness of mass rearing food on the establishment of C. pallens in banker plant system under fluctuating temperatures conditions. The real effect of temperature on the characteristics of the BCAs is the closest to the daily temperature fluctuations that occur in the field. We didn’t have to overgeneralize the significances here, and we made several modifications in the Discussion based on your suggestions to make clearer the central topic. And we also had discussed the other factors that would account for the variations. “Fluctuating temperature, higher humidity, photoperiod, and light intensity may be responsible for the lower survival rate, shorter longevity, and even lower reproduction of predators in field. We suggest that temperature is a key variable that restricts adult longevity, fecundity, and predation rate of C. pallens in field.”

Overall, I was excited to see the results of the paper after reading the abstract, but I found it hard to extract key messages useful to policymakers and professionals, probably in large part due to the lack of connection with other published work and need for improved structure of the current manuscript.

Response: We appreciate your careful comments in our manuscript. We had rephrased the discussion and added more details about the connection with other published works (life table, Mu et al. 1980, Zhao 1988, Yu et al. 2013, Zhao et al. 2015; predation rate, El-Serafi et al. 2000, Cheng et al. 2015). And other works about the temperatures on development rates of insects are also cited to improve the discussion (McCalla et al. 2019, Milosavljevi´c et al. 2019, 2020, Chen et al. 2021). Please see more details in the revision.

Round 2

Reviewer 1 Report

#insects-1406605

Title: The fitness of mass rearing food on the establishment of Chrysopa pallens in banker plant system under fluctuating temperatures conditions

Brief.

The manuscript has been improved after revisions. Nonetheless, some issues still require attention in the manuscript mainly in the results section. In general comments were left suggestions and corrections.

General comments

L97 and 368 – the abbreviation of Integrated pest management was suggested at the beginning of the introduction section. Thus, it should be used in the MS. Please, ‘integrated pest management strategies should be changed to IPM strategies.

L162 – Please, the scientific name of the species should be written fully at the beginning of the sentence. Please, it should be checked in the entire manuscript.

Tables 1, 2, and 3 - There are doubts about the letter 'a' in the table indicating that there was no difference between treatments if the same information appears in the P-value column. The notes are unnecessary if there were no different letters in the tables. In addition, the information ‘paired bootstrap test, B = 100 000, P < 0.05’ should be moved to the legend of the tables.

L313 to 315 - Was the statistical analysis performed comparing females and males of C. pallens? Although the difference in aphid consumption by C. pallens was highest in females than males, the statement cannot conclude that there was a difference. Please rephrase the sentence with correct information.

L449 – Please, Corcyra cephalonica should be abbreviated in the sentence.

Author Response

Dear academic editor and reviewers,

Thank you for your works on ‘The fitness of mass rearing food on the establishment of Chrysopa pallens in banker plant system under fluctuating temperatures conditions’ (Manuscript ID: insects-1406605). We appreciate your careful feedback and comments in our manuscript. We believe that the manuscript improved based on your contributions. We have taken Reviewers' comments into consideration word by word. Please, see our point-by-point responses to your comments and suggestions. More details are provided in the revision with “Track Changes”.

Thank you for considering our revision notes.

Do not hesitate to contact us if you have any questions regarding the revision of our manuscript. We look forward to hearing from you soon.

Best regard!

Su Wang

BAAFS, Beijing, China

wangsu@ipepbaafs.cn

Reviewer 1

Title: The fitness of mass rearing food on the establishment of Chrysopa pallens in banker plant system under fluctuating temperatures conditions

Brief.

The manuscript has been improved after revisions. Nonetheless, some issues still require attention in the manuscript mainly in the results section. In general comments were left suggestions and corrections.

General comments

L97 and 368 – the abbreviation of Integrated pest management was suggested at the beginning of the introduction section. Thus, it should be used in the MS. Please, ‘integrated pest management strategies should be changed to IPM strategies.

Response: The reviewer is right. Done as requested. Thanks.

L162 – Please, the scientific name of the species should be written fully at the beginning of the sentence. Please, it should be checked in the entire manuscript.

Response: Thanks. We had corrected “C. pallens” into “Chrysopa pallens”.

Tables 1, 2, and 3 - There are doubts about the letter 'a' in the table indicating that there was no difference between treatments if the same information appears in the P-value column. The notes are unnecessary if there were no different letters in the tables. In addition, the information ‘paired bootstrap test, B = 100 000, P < 0.05’ should be moved to the legend of the tables.

Response: The reviewer is right. We had done as requested. Thanks.

L313 to 315 - Was the statistical analysis performed comparing females and males of C. pallens? Although the difference in aphid consumption by C. pallens was highest in females than males, the statement cannot conclude that there was a difference. Please rephrase the sentence with correct information.

Response: Thanks. We didn’t perform the statistical analysis comparing females and males of C. pallens. So, we rephrased the sentence into “A male adult could consume as many as 270 aphids, while a female adult could consume 609 aphids in the C. cephalonica egg treatment under fluctuating temperature conditions in the greenhouse.”

L449 – Please, Corcyra cephalonica should be abbreviated in the sentence.

Response: Thanks. We had changed “Corcyra cephalonica” into “C. cephalonica”.

Reviewer 2 Report

Dear Authors,

Thank you for your point by point clarification. The manuscript has been greatly improved.

Some edits and notes:

Line 49: Add space between 'A. craccivora' and 'in'.

Line 52-382: This line came from line 379-382 and needs rewording. You can start by "Although there is no significance differences, this xxx food is better than that xx in terms of xxx'.

Line 59: Delete 'IPM programs'.

Line 451: 'eggs do not move' correct. May replace 'do not move' by 'stationary'.

Thanks for hard work.

Author Response

Dear academic editor and reviewers,

Thank you for your works on ‘The fitness of mass rearing food on the establishment of Chrysopa pallens in banker plant system under fluctuating temperatures conditions’ (Manuscript ID: insects-1406605). We appreciate your careful feedback and comments in our manuscript. We believe that the manuscript improved based on your contributions. We have taken Reviewers' comments into consideration word by word. Please, see our point-by-point responses to your comments and suggestions. More details are provided in the revision with “Track Changes”.

Thank you for considering our revision notes.

Do not hesitate to contact us if you have any questions regarding the revision of our manuscript. We look forward to hearing from you soon.

Best regard!

Su Wang

BAAFS, Beijing, China

wangsu@ipepbaafs.cn

Reviewer 2

Dear Authors,

Thank you for your point-by-point clarification. The manuscript has been greatly improved.

Some edits and notes:

Line 49: Add space between 'A. craccivora' and 'in'.

Response: Thanks. Done as requested.

Line 52-382: This line came from line 379-382 and needs rewording. You can start by "Although there is no significance differences, this xxx food is better than that xx in terms of xxx'.

Response: Thanks. We are focused on the fitness of mass rearing food C. cephalonica egg on the establishment of C. pallens compared to aphid treatment. And the C. cephalonica egg display a similar efficacy to C. pallens as aphid. So, we mortified the sentence “The use of Corcyra cephalonica eggs as a food for mass rearing did not affect the population colonization of C. pallens in a Vicia faba–A. craccivora banker plant system compared to aphid treatment” into “Corcyra cephalonica eggs didn’t significantly weaken the performances of C. pallens in a Vicia fabaA. craccivora banker plant system compared to aphids” We hope it is clearer now.

Line 59: Delete 'IPM programs'.

Response: Thanks. Done as requested.

Line 451: 'eggs do not move' correct. May replace 'do not move' by 'stationary'.

Response: Thanks. we had replaced 'do not move' by 'were stationary”.

Reviewer 3 Report

The authors have done a nice job addressing all of my original comments and suggestions, and those of other reviewers. I have no further comments to improve the paper. Thank you and good luck.

Author Response

Dear academic editor and reviewers,

Thank you for your works on ‘The fitness of mass rearing food on the establishment of Chrysopa pallens in banker plant system under fluctuating temperatures conditions’ (Manuscript ID: insects-1406605). We appreciate your careful feedback and comments in our manuscript. We believe that the manuscript improved based on your contributions. 

Thank you for considering our revision notes.

Do not hesitate to contact us if you have any questions regarding the revision of our manuscript. We look forward to hearing from you soon.

Best regard!

Su Wang

BAAFS, Beijing, China

wangsu@ipepbaafs.cn

This manuscript is a resubmission of an earlier submission. The following is a list of the peer review reports and author responses from that submission.

Round 1

Reviewer 1 Report

The stated aim of this study is to evaluate the impact of mass rearing food on biocontrol services provided by Chrysopa pallens. The study is conducted in a greenhouse but the larvae are kept isolated inside test tubes and supplied with aphids. The method seems to me totally inadequate to evaluate the biocontrol service. To assess the biocontrol service, predators must be able to actively search for prey on plants. Instead of the biocontrol service, life history traits were evaluated in fluctuating temperature conditions.

Since the experiment starts at the egg stage, it is reasonable to assume that the authors expect fitness changes due to selection during mass rearing. However, it is not explained why they expect selection would affect the ability to use the aphid as food and not, for example, the ability to forage. Is there any reference to support this choice? It seems that the authors are interested in the application of a statistical technique rather than in the biological phenomenon.

Reviewer 2 Report

INSECTS-1345953

Title: Impact of mass rearing food on biocontrol services provided by Chrysopa pallens under fluctuating temperatures conditions

Brief.

The manuscript covers an important topic of the interactive effects of fluctuating temperatures on the natural enemy fed on two species of prey, perhaps offering a new perspective. However, the manuscript is preliminary and therefore not currently suitable for publication. The critical point was found in the discussion section which should be enhanced with robust conclusions supported by the present results and references. The replication of results in the discussion section is not interesting and leaves the discussion weak as well as the conclusion. General comments provide suggestions in trying to improve the manuscript.

General comments

L101 – There were no variations in the relative humidity of the air in the place where C. cephalonica was raised?

L309, L363, and 364 – Please, the author, order, and family of species should be provided in the sentence to pattern the manuscript.

L312 – What are the data published above? Perhaps, the information “given above” should be deleted from the sentence.

L329-330 - Temperature has a great impact on the biological parameters of arthropods as is already known. Perhaps, in addition to the suggestion about temperature, the results could be better explored with the introduction of more information about the results found in this study.

L341 – Was the temperature fluctuation the only factor due to the weak resistance of C. pallens to adversity. What about relative humidity, solar radiation, etc.?

L359 – Why was used ‘however’ in the sentence?

L359 to 370 – The subheading ‘4.2. Impact of Mass Rearing Food on the Biological Performance of Chrysopa pallens’ has a weak discussion about the results found on the C. pallens fed on M. japonica and C. cephalonica eggs. In the first paragraph, no conclusions were provided, and the results described in the results section were replicated in the discussion section. In the second paragraph, further studies on the behavior of two species were suggested, and it was concluded C. cephalonica eggs are suitable for mass rearing of the C. pallens. Please, the subheading 4.2 should be better discussed with information to support the conclusion.

L366 to 368 – The sentence is confusing, and it should be shorted and clearer.

L374 – Please, avoid an abbreviation at the beginning of a sentence, as well as the contract form of the English writing in scientific research.

Reviewer 3 Report

Comments attached.
